# HEXA: SELF-IMPROVING FOR KNOWLEDGE-GROUNDED DIALOGUE SYSTEM

## ABSTRACT

A common practice in knowledge-grounded dialogue generation is to explicitly utilize intermediate steps (e.g., web-search, memory retrieval) with modular approaches. However, data for such steps are often inaccessible compared to those of dialogue responses as they are unobservable in an ordinary dialogue. To fill in the absence of these data, we develop a self-improving method to improve the generative performances of intermediate steps without the ground truth data. In particular, we propose a novel bootstrapping scheme with a guided prompt and a modified loss function to enhance the diversity of appropriate self-generated responses. Through experiments on various benchmark datasets, we empirically demonstrate that our method successfully leverages a self-improving mechanism in generating intermediate and final responses and improves the performances on the task of knowledge-grounded dialogue generation.

## 1 INTRODUCTION

Along with the progress of Language Model (LM) pretraining, open-domain dialogue models have evolved to leverage the advantage of the transformer architecture's generalization ability (Zhang et al., 2019; Freitas et al., 2020; Roller et al., 2021; Xu et al., 2022a; Shuster et al., 2022b; Thoppilan et al., 2022). While model scaling also improves the dialogue quality (Freitas et al., 2020) as seen in large LMs, relying on sole LMs casts limitations such as hallucination and the lack of faithfulness by outdated training data (Brown et al., 2020; Thoppilan et al., 2022; Chowdhery et al., 2022). In order to overcome the limitations, prior works have adopted a modular design where multiple modules generate intermediate texts (e.g., to retrieve documents) before the final response (Lewis et al., 2020; Adolphs et al., 2021; Zhang et al., 2021; Shuster et al., 2022a). Furthermore, recent works have taken the modular design to dialogue models (Dinan et al., 2019; Lian et al., 2019; Zhao et al., 2020; Komeili et al., 2022; Huang et al., 2021; Xu et al., 2022a; Shuster et al., 2022b; Thoppilan et al., 2022). Among them, Komeili et al. (2022); Shuster et al. (2022b) have shown promising results in dialogue generation. Specifically, they adopted a modular design to integrate external knowledge (e.g., internet) and internal knowledge (e.g., memory) in dialogue models. For example, in Komeili et al. (2022), a LM first decides whether to access a knowledge in a form of text generation. Upon deciding to access knowledge, the LM generates an appropriate query for knowledge retrieval from external sources such as search engines. Then, the LM generates a response based on extracted knowledge from the accessed data. See Figure 2 of Appendix B for an illustrative example.

Regarding each intermediate phase as a separate module, a convenient method of training these modules would be to apply supervised learning on each module using individual datasets (Dinan et al., 2019; Shuster et al., 2022a; Glass et al., 2022; Shuster et al., 2022b). However, as the final response can be inferred only after multiple intermediate steps have been generated, there exists multi-depth dependencies between modules, which hinders the modules to be well learned from independent supervised training. Moreover, most off-the-shelf datasets are not well-aligned with the modules as they were not originally designed for them. Therefore, it is inevitable that there is a discrepancy between training and inference, possibly leading to severe performance degradation. To avoid the discrepancy, incorporating the intermediate steps into training is necessary and one approach is to treat them as latent variables (Lewis et al., 2020; Zhang et al., 2021; Lian et al., 2019; Zhou et al., 2020; Zelikman et al., 2022). With the latent variable model, we propose a novel self-improving method for dialogue models that use bootstrapped samples from both the latent variable and the final response. In our method, the model can use not only the self-generated samples to form an empirical target distribution but also samples more plausible response candidates

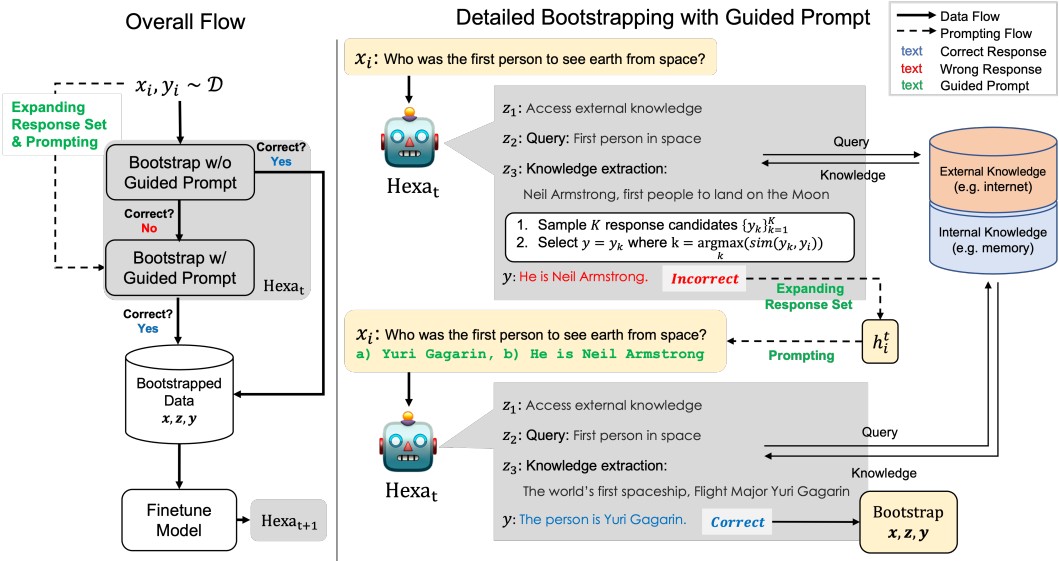

Figure 1: Schematic diagram of Hexa at iteration $t$. **(Left)** The overall flow of data bootstrapping and finetuning in Hexa. Given a dialogue context and response pair $(x_i, y_i)$ sampled from the dataset $\mathcal{D}$, Hexa runs through bootstrapping phases represented in the gray shaded area. The model is then finetuned on the bootstrapped data and the process repeats. **(Right)** More detailed sketch of Hexa. With input $x_i$ the model generates intermediate steps, $z_1, z_2$, and $z_3$, and a response $y$. **(Right-Top)** Due to the mis-informed intermediate step, $y$ (red) is rejected by the matching function and is added to response set $h_i^t$. **(Right-Bottom)** The model generates a response again with a guided prompt, highlighted in green below $x_i$. This time, $z_3$ is well aligned with $x_i$, leading a correct response. Then the sample ($x_i$, **z**, and $y$) (blue) is stored in bootstrapped data on which the model is finetuned.

guided by additional information. Furthermore, we propose a guided prompting scheme based on previous generated responses to provide more information for the model in producing a proper response. A visual depiction of the self-improving and bootstrapping scheme, which we name *Hexa*[1], is shown in Figure 1. Through empirical analysis, we show that our method consistently improves the dialogue generation capabilities of the base model and outperforms an existing self-improving method in various categories of dialogue generation tasks: Knowledge intensive question answering, Knowledge-grounded dialogue, Open-domain dialogue, and Task-oriented dialogue.

To summarize, our main contributions are: (1) a self-improving modular method for knowledge-grounded dialogue generation, which enables proper learning of intermediate modules in absence of their ground-truth data; (2) a novel bootstrapping scheme with a guided prompt and a modified loss function for diverse and appropriate generation of intermediate and final responses to be self-trained; and (3) an empirical validation of Hexa that outperforms the previous supervised learning and self-improving methods and moreover demonstrates the effectiveness on various dialogue tasks for the modular-based, knowledge-grounded dialogue system.

## 2 RELATED WORKS

**Open-Domain Dialogue Systems** Alongside the growth of deep learning-based LMs, open-domain dialogue systems have also adopted similar approaches leading to fully end-to-end deep learning models for human-like dialogue generation. DialoGPT (Zhang et al., 2019) and Meena (Freitas et al., 2020) are well known examples of single module approach to end-to-end neural dialogue generation models. Although these models can produce fluent open-domain dialogue and the quality can improve by model scaling, the factual grounding of the generated responses has not seen much improvement. To alleviate this issue, several works have proposed knowledge-grounded generative models consisting of a document retriever and response generator (Lewis et al., 2020; Adolphs et al.,

---

[1]We named it from the abbreviation of our title: **S**elf-**I**mproving for **K**nowledge-Grounded Dialogue **S**ystem (SIKS), which is the phonetic symbols of 6.

2021; Zhang et al., 2021; Shuster et al., 2022a). Lewis et al. (2020) treated the retrieved documents as a single latent variable and propose jointly training the retriever and generator with a fixed document encoder for knowledge intensive natural language tasks. Shuster et al. (2022a) proposed a modular model consisting of three sequentially invoked modules for search query, knowledge response, and response generation. Along with the progress of such studies, a series of recent works (Roller et al., 2021; Komeili et al., 2022; Xu et al., 2022a; Thoppilan et al., 2022; Shuster et al., 2022b) have shown a promising direction of systematically producing the response using multiple modules and inspired a development of large-scale open source dialogue model (Schuhmann, 2022), based on the systematic approach.

**Self-Improving for LMs** Although the modular design is promising, obtaining the gold label data for intermediate outputs of the modules is expensive and difficult, especially in open-domain dialogue. In order to overcome this issue, one can consider data-augmentation approach. Data-augmentation methods often leverage inductive bias such as delexicalisation (Hou et al., 2018), human annotation (Sun et al., 2021), and predefined operations for automatic augmentation (Niu & Bansal, 2019) to improve synthetic data quality. Zhang et al. (2020) augmented paired dialogue data by matching an additional unpaired sample and paired sample retrieved according to a predefined similarity score. Several recent works have also shown that LMs can be improved from self-generated data from few-shot prompting. Wang et al. (2022a) proposed an automatic instruction data generation by prompting from a small set of seed human-written instructions and model-generated instructions. The generated instructions are filtered by a simple word overlap function and then used to train the LM with diverse prefixes. The updated LM from the synthetic instructions shows comparable results to the updated LM from human collected instructions (Wang et al., 2022b). Zelikman et al. (2022) provided an iterative self-improving method, referred to as STaR, that leverages a small number of intermediate steps such as *chain-of-thoughts* (Wei et al., 2022) or *scratchpads* (Nye et al., 2021) examples to bootstrap reasoning experiences that might guide the improvements.

Here, it is noted that in comparison to STaR (Zelikman et al., 2022), our proposed method has core differences in three aspects that make our method appropriate for self-improving in knowledge-grounded dialogue generation. First, due to the modular generation of knowledge-grounded dialogue, in contrast to the implicit reasoning steps in STaR, we explicitly generate intermediate knowledge extraction steps by separate modules. Therefore, second, while STaR includes the ground truth response only as a hint in the prompt, we incorporate a set of past responses in our guided prompt to prevent the intermediate steps from being collapsed to the ground truth response. Last, to account for one-to-many relationship between dialogue context and correct responses, we modify the matching function to cover multiple response candidates and provide diverse label candidates to the intermediate steps, while enriching the learning signals.

## 3 KNOWLEDGE GROUNDED DIALOGUE SYSTEM

Motivated by Shuster et al. (2022b), Hexa uses internal and external knowledge such as long-term memory and retrieved documents from search engine are integrated in the process of generating the dialogue response. This is a latent variable model in which $p(y|x; \theta) = \sum_z p(y|x, z; \theta)p(z|x; \theta)$, where $x \in X$ and $y \in Y$ are dialogue context and a response from some data distribution i.e., $x, y \sim p_{\mathcal{D}}(X, Y)$, $z \in Z$ represents the intermediate products of external and internal knowledge and $\theta$ is the set of model parameters. The set of latent variables specific to Hexa is shown in Figure 3 of Appendix B.

Given a distribution of desired dialogue context and respective responses that we wish to train our model on, denoted $p_{\mathcal{D}}$, our primary objective is to find a set of model parameters $\theta$, that maximizes the expected conditional probability as

$$J(\theta) = \mathbb{E}_{x,y \sim p_{\mathcal{D}}} p(y|x; \theta). \tag{1}$$

Approximating the data distribution $p_{\mathcal{D}}$ with finite samples, $\mathcal{D} = \{x_i, y_i\}_{i=1}^{|\mathcal{D}|}$ and substituting the latent variable model, the objective becomes

$$J(\theta) = \sum_{x,y} p_{\mathcal{D}}(x)p_{\mathcal{D}}(y|x)p(y|x; \theta)$$
$$\approx \frac{1}{|\mathcal{D}|} \sum_i \sum_{y,z} p_{\mathcal{D}}(y|x_i)p(y, z|x_i; \theta) = \frac{1}{|\mathcal{D}|} \sum_i \mathbb{E}_{y,z \sim p(\cdot|x_i; \theta)} \mathbb{1}(y = y_i). \tag{2}$$

Using the log trick of policy gradient, the gradient of the objective function is

$$
\begin{aligned}
\nabla_\theta J(\theta) &\approx \frac{1}{|\mathcal{D}|} \sum_i \mathbb{E}_{y,z \sim p(\cdot|x_i;\theta)} \mathbb{1}(y = y_i) \nabla_\theta \log p(y, z|x_i; \theta) \\
&= \frac{1}{|\mathcal{D}|} \sum_i \mathbb{E}_{y,z \sim p(\cdot|x_i;\theta)} \mathbb{1}(y = y_i) \nabla_\theta [\log p(y|z, x_i; \theta) + \log p(z|x_i; \theta)].
\end{aligned}
\tag{3}
$$

In this form, we can conveniently calculate the sample-based approximation of the gradient using samples of each log probability of the intermediate step and the final response.

## 4 HEXA: SELF-IMPROVING IN KNOWLEDGE GROUNDED DIALOGUE SYSTEM

In Hexa, the model is trained for several iterations, where each iteration composes of bootstrapping and finetuning phases. During the bootstrapping phase, the model will collect self-generated samples according to a matching function. Then, the method proceeds to finetune the model with the bootstrapped dataset (See Figure 1 and Algorithm1 of Appendix A). Since a dialogue response generation is a *one-to-many* situation with possibly long sentences as answers, the indicator function in Equation 3 with exact match has extremely a low chance of producing a useful learning signal. To alleviate this issue, we change the indicator function from a dirac delta function to $B(y, y_i) = \mathbb{1}(\text{sim}(y, y_i) > b)$ where $\text{sim}(\cdot, \cdot)$ is a similarity function (e.g., BLEU (Papineni et al., 2002) or ROUGE (Lin, 2004)) and $b$ is a hyperparameter threshold.

A core difference that distinguishes Hexa from STaR is in formulation of the guided prompt used when the model falsely predicts the final response. Instead of including the ground truth only, we propose to add previous unmatched responses along with the ground truth to compose the guided prompt. More formally, we let the response set for the guided prompt at iteration $t$ to be defined as $h_i^t = \{y_i^j\}_{j=1}^t \cup y_i$ where each $y_i^j$ is *unmatched* response sampled from the model $p_{\theta, j \leq t}$ in earlier iterations $j \leq t$. Then, during the bootstrapping phase, when the model generates unmatched response, Hexa augments previously gathered unmatched response set along with the ground truth to the input prompt in an *Alphabetical List* (AL) with random order. An example of the guided prompt shown in the right-bottom part (highlighted in green below $x_i$) of Figure 1 presents an AL with the ground truth (Yuri Gagarin) and the unmatched response (He is Neil Armstrong) for given $x_i$. The unmatched response is added to the response set whenever the prediction fails. Therefore, during one instance of bootstrapping, if the model fails both before and after the augmentation of guided prompt, both responses will be added to the response set. The unmatched response set contains recent unmatched predictions up to $H$. The study on different values of $H$ is covered in subsection 5.6.

There are three intuitions behind formulating the guided prompt of Hexa using a combined set of unmatched responses and the ground truth label. First, a LM has a tendency of referencing the prompt. If the guided prompt only includes the ground truth label of the final response then the model is vulnerable to simply copying the ground truth response throughout the intermediate steps. Then, the responses of the intermediate steps would have deviated from their intentions, and furthermore the model may collapse to simply copying the guided prompt regardless of the dialogue context. To prevent such phenomenon, we augment the guided prompt with responses other than the ground truth itself. Second, the purpose of adding previous unmatched responses and not just a random response is because we want the responses in the guided prompt to work as a collection of signals that directs the input prompt to a more easily answerable space, by providing more information. In addition, as the model continuously improves from the bootstrapped sets, the unmatched responses eventually become counterfactual examples, being different from the ground truth while having potentially helpful and relevant information. Lastly, the purpose of having the ground truth in the guided prompt is to serve as a correction term to the signals created by the unmatched responses as they are not always guaranteed to provide relevant information. Note that the unmatched responses may or may not be relevant information but the guided prompt which contains the unmatched responses in an AL imposes autonomy to the model to interpret it as a multiple choice or a list of relevant information. We show empirical observations of these intuitions in more detail in subsection 5.6.

Once the bootstrapping stage is over, the model is finetuned on the collected bootstrapped data. Gathering all modifications, the final objective of Hexa at iteration $t$ can be described as

$$
\nabla_\theta J^{\text{Hexa}}(\theta) = \frac{1}{|\mathcal{D}|} \sum_i \mathbb{E}_{y,z \sim p(\cdot|x_i, h_i^t; \theta)} B(y, y_i) \nabla_\theta \log p(y, z|x_i; \theta).
\tag{4}
$$

Table 1: Average performance for each task: question answering (QA), knowledge-grounded dialogue (KGD), open-domain dialogue (ODD), and task-oriented dialogue (TOD). The average results across all tasks are shown in the columns under *Average*.

| Model | QA F1 | QA R-L | KGD F1 | KGD R-L | ODD F1 | ODD R-L | TOD F1 | TOD R-L | *Average* F1 | *Average* R-L |
|---|---|---|---|---|---|---|---|---|---|---|
| BB3 | 21.36 | 34.38 | 15.71 | 14.03 | 18.42 | 15.15 | 15.96 | 14.71 | 17.83 | 19.48 |
| BB3-60K | 20.33 | 34.23 | 15.59 | 13.98 | 18.6 | 15.38 | 16.08 | 14.76 | 17.59 | 19.49 |
| BB3-SL | **24.21** | 35.32 | 16.06 | 14.37 | 18.77 | 15.49 | 16.84 | 15.54 | 18.87 | 20.03 |
| STaR | 22.33 | 34.54 | 16.93 | 15.27 | 19.86 | 17.08 | 18.86 | 17.64 | 19.25 | 20.84 |
| Hexa | 22.65 | **36.34** | **19.62** | **17.15** | 19.62 | 17.15 | **20.22** | **18.34** | **20.83** | **22.25** |

**Response Generation** To make sure the predicted responses at each iteration of the bootstrap are well aligned with the ground truth, we sample $K$ different responses and select a response that has highest similarity score with the ground truth. For the generation method, we may consider search-based methods such as *beam* search for enhanced similarity, however, we stay with the stochastic sampling as search-based methods restrict the diversity in generation (Freitas et al., 2020). The study on different values of $K$ and the response selection standards are covered in subsection 5.6.

## 5 EXPERIMENTS

### 5.1 EXPERIMENT SETUP

**Model** We use 3B version of BB3 (Shuster et al., 2022b). This version is based on the encoder-decoder transformer LM that is pretrained from large scale dialogue datasets (Shuster et al., 2022a).

**Baselines** In order to measure the effect of each component proposed in Hexa, we choose BB3 for a baseline since Hexa can be thought of as finetuned version of BB3. Since the self-improving method contains additional finetuning on the bootstrapped data, we also compare Hexa to a further finetuned version of BB3, which is denoted as *BB3-60K*, for a fair assessment. In BB3-60K, the model is supervised trained for 60K steps instead of the original 30K in BB3, using the same pretraining dataset. Moreover, we add another version of BB3, which is supervised trained on the same dataset used for self-improving methods, to the baselines and it is denoted as *BB3-SL*. We also include a modified version of STaR (Zelikman et al., 2022), where only the ground truth label is included in the prompt for bootstrapping and the model generates the same intermediate responses as Hexa instead of the rationales. The modification was inevitable since the original form of STaR was not design for the modular framework as in our setting. Furthermore, our implementation of STaR uses the same similarity based matching function as Hexa instead of the exact matching function used in the original STaR, as it is unsuitable for bootstrapping dialogue data.

**Dataset** We experiment on various dialogue generation tasks including Question Answering (QA), Knowledge-Grounded Dialogue (KGD), Open-Domain Dialogue (ODD), and Task-Oriented Dialogue (TOD). As in Shuster et al. (2022b), for QA task evaluation, we use MS Marco (Nguyen et al., 2016) and TriviaQA (Joshi et al., 2017). For the KGD, we use Wizard of Wikipedia (WoW)(Dinan et al., 2019), Wizard of Internet (WoI) (Komeili et al., 2022), and Feedback on Interactive Talk & Search (FITS) (Xu et al., 2022b). In these QA and KGD tasks, we test the abilities such as factual generation and external knowledge utilization of the models. Moreover, GoogleSGD (Lee et al., 2022) is used as a TOD dataset to evaluate the transferability of the algorithm to other tasks. Finally, PersonaChat (Zhang et al., 2018) and Multi-Session Chat (Xu et al., 2022a) are used as ODD task evaluation for general ability of dialogue generation including internal knowledge utilization. In our experiments, these eight tasks are used during finetuning of the corresponding models (i.e. BB3-SL, and variants of STaR and Hexa) and categorized as *Seen* in the experimental results.

Furthermore, to test unseen task generalization, we pick three datasets from different categories: Funpedia (Dinan et al., 2020) for KGD, Empathetic Dialogues (ED) (Rashkin et al., 2019) for ODD, and Taskmaster (Byrne et al., 2019) for TOD. In the experimental results, the evaluations for these three tasks are categorized as *Unseen*.

### 5.2 IMPLEMENTATION DETAILS

Here, we describe the details of implementation and training. We use the same settings for the baselines and Hexa unless stated otherwise. For the external knowledge source, we retrieve relevant

Table 2: Results for tasks unseen during finetuning. The average results across all tasks are shown in the columns under *Average*. BB3-60K result is added for reference.

|  | Funpedia | | ED | | Taskmaster | | *Average* | |
|  | Dinan et al. (2020) | | Rashkin et al. (2019) | | Byrne et al. (2019) | | | |
| Model | F1 | R-L | F1 | R-L | F1 | R-L | F1 | R-L |
|---|---|---|---|---|---|---|---|---|
| BB3-60K | 16.42 | 15.12 | 17.06 | 14.7 | 14.41 | 13.1 | 15.96 | 14.31 |
| BB3-SL | 16.88 | 15.21 | 16.24 | 13.97 | 13.08 | 12.17 | 15.4 | 13.78 |
| STaR | 17.58 | **16.36** | 17.37 | 15.38 | 16.4 | 15.03 | 16.4 | 15.03 |
| Hexa | **18.08** | 16.22 | **19.62** | **17.38** | **17.94** | **16.28** | **18.55** | **16.63** |

Table 3: Human evaluation results on KGD, ODD, and TOD testsets. A pairwise t-test is conducted to verify statistical significance of the improvements, and the corresponding results in bold are significantly better than those from the baseline model (***: $p < 0.001$, **: $p < 0.01$, *: $p < 0.05$).

|  | Fluency (%) | | Relevance (%) | | Faithfulness (%) | |
| Model | Seen | Unseen | Seen | Unseen | Seen | Unseen |
|---|---|---|---|---|---|---|
| STaR | 69.92 | 73 | 58.92 | 59.5 | 48.75 | 53.17 |
| Hexa | **86.33***** | **86.67***** | **73.42***** | **70.67**** | **66.5***** | **64*** |

documents using BM25 based search engine built on the wikipedia corpus (Karpukhin et al., 2020). During the self-improving process, we linearly increase the number of bootstrapped samples by $10\%$, starting from $4,000$ samples in the initial iteration. The learning rate is fixed as $2e{-}6$ and the model is finetuned using four A100 gpus with batch size of 1 per gpu and gradient accumulation of 4, yielding total batch size of 16.

The datasets used to train our base model (BB3) have target outputs of various lengths, and our model is likely to generate longer and more natural dialogue responses even for short question answering tasks such as TriviaQA. Since metrics that focus on measuring a precision such as BLEU would not be suitable for taking account into such flexibility in answering, and we choose the most popular recall-oriented metric for measuring language generation, ROUGE-L, as our similarity measure. We include further study on Sentence-BERT (Reimers & Gurevych, 2019) as the similarity measure of Hexa in Appendix G.

Choosing the appropriate threshold value $b$ of matching function $B$ in Equation 4 is also important as too low threshold can include undesired target responses in the bootstrapped data, leading to performance degradation upon finetuning, and a too high threshold can overly limit the number of bootstrapped instances and overfit the model to a narrow set of responses. For these reasons, we need to find good threshold values efficiently without repeatedly training the model, which we achieve by heuristic search on each task. Specifically, we run a single training iteration multiple times on each task with an initial threshold value of 0.2 that is increased by 0.05 after each iteration. The process repeats until there is no improvement on the model's performance on a small subset of validation dataset and the last threshold value is selected for the task. The final threshold values and ablation study regarding to the threshold variation are provided in Appendix G.

## 5.3 Automatic Evaluation

To measure the quality of generated responses from different models, we utilize the classical overlap-based metrics: F1 score and ROUGE-L (R-L) (Lin, 2004) to measure similarity from the ground truth on the test datasets. The response is generated in *End-to-End* manner. Namely, we do not use the gold labels for the intermediate steps. The results for each task are shown in Table 1. For the generalization capabilty, we compare the results of same metrics on a set of unseen tasks that are not included during the Hexa training. The results of unseen experiment is shown in Table 2. The results of both seen and unseen tasks show that Hexa achieves the highest overall scores in all the metrics. The performance increase in Hexa compared to BB3 and BB3-SL indicates that training intermediate steps without their ground truth data still leads to improvement, possibly better than supervised training with fixed labels.

Furthermore, in Table 1 the score gap in (STaR - BB3) is 1.42 and 1.36 for F1 and R-L respectively. On the other hand, the score gap in (Hexa - STaR) is 1.58 and 1.41 for F1 and R-L. While Hexa is also trained from BB3 as STaR, the score gap is even greater in (Hexa - STaR) than (STaR -

Table 4: Generated examples from **(Left)** WoI and **(Right)** WoW. Here, all the models access the external knowledge with generated query by the models. The response is generated based on the prior step (knowledge). These examples display STaR's tendency to copy the information contained in knowledge when generating the response, while Hexa stays on generating response for the original input, using the knowledge. More samples can be found in Appendix K.

| Input | I cannot remember who was the second president. |
|---|---|
| query | second president of united states |
| knowledge | President pro tempore of the United States The Second Lady of the United States (SLOTUS) is the [...] |
| **BB3-SL** | The second lady of the united states was jennie tuttle hobart. |
| query | George washington |
| knowledge | George steptoe washington (August 17, 1771 - January 10, 1809) was a planter, militia officer and nephew of the first president of the united states' |
| **STaR** | George washington. |
| query | second president |
| knowledge | The second president of the united states was adams. |
| **Hexa** | The second president is adams. |

| Input | How do you make a jerk chicken marinade? |
|---|---|
| query | Jerk chicken recipe |
| knowledge | "Poulet au fromage"_0 / "along with hard dough bread, deep fried ""cassava bammy"" (flatbread, usually with fish), Jamaican fried dumplings (known as ""Johnny-cake"" or ""journey cakes"" |
| **BB3-SL** | You can make jerk chicken by marinating the chicken in jerk seasoning. |
| query | Jerk chicken recipe |
| knowledge | Other ingredients may include cloves, cinnamon, scallions, nutmeg, thyme, garlic, brown sugar, ginger, and salt. |
| **STaR** | Other ingredients include cloves and cinnamon. |
| query | Jerk chicken recipe |
| knowledge | Other ingredients may include cloves, cinnamon, scallions, nutmeg, thyme, garlic, brown sugar, ginger, and salt. |
| **Hexa** | You need to marinate the chicken in a mixture of spices and sugar. |

BB3), indicating the difference and significance of improvement from the guided prompt with a set of bootstrapped responses is a critical factor in the improvement. As we noted in section 4, we hypothesize that guided prompt with just the ground truth response is biased and easy to collapse to simply copying the guided prompt throughout the intermediate steps. We show two qualitative examples of such cases in Table 4, where the response generated by STaR is a mere copy of the knowledge, regardless of the original question asked in the dialogue context. On the other hand, Hexa successfully answers the original question using the knowledge obtained from the intermediate step in both examples. We further analyze the copying issue and empirically show that Hexa alleviates the issue in subsection D.1 of Appendix D.

We include further automatic evaluation results as follows: (1) module-wise evaluation of intermediate steps results (see Appendix C); (2) comparison to assess effectiveness of bootstrapping (see Appendix D); (3) robustness evaluation on totally-unseen tasks[2] of OpenDialKG (Moon et al., 2019) (see Appendix E); and (4) evaluation models without modules (see Appendix H).

## 5.4 HUMAN EVALUATION

Following Rashkin et al. (2021), we use human evaluation to compare the generated responses from Hexa with previous self-improving method STaR (Zelikman et al., 2022) for a comprehensive evaluation. The feedbacks were collected from 10 human experts and we asked them to evaluate the responses in terms of three qualities: *Fluency*, *Relevance*, and *Faithfulness*. *Fluency* evaluates whether the response is understandable, self-consistent, and proficient. *Relevance* assesses whether the generated knowledge and the corresponding response are appropriate to the dialogue history. *Faithfulness* measures whether the response is supported by the knowledge and the dialogue context. In total, 180 data samples are randomly selected from the testsets. Specifically, 20 samples are randomly selected from 9 different tasks of KGD, ODD and TOD. The qualities of the responses are measured by A/B testing on the three aspects, which reflects whether the model generates an equally good or better response than the other. Further details and annotator instructions are included in Appendix I. As shown in Table 3, Hexa significantly outperforms the baseline in all three categories. It is noted that, during human evaluation, the annotators are asked to compare Relevance of knowledge generated from either search, entity, or memory knowledge modules along with the response. The preference of Hexa shown in Table 3 implicitly indicates the better performance of the knowledge module of Hexa.

## 5.5 DIVERSITY OF RESPONSES

Here, we conduct an automatic evaluation for diversity between final responses. There exists a tradeoff between the diversity and correctness, as group of correct answers would tend to resemble

---

[2]The tasks which are not used during both BB3-training and finetuning.

each other compared to set of random answers. Therefore we specially design a method to measure the appropriate diversity within a certain boundary of correctness. We first randomly sample intermediate steps z and y 10 times for each instances. Then, we select samples that satisfy the matching function. Furthermore, we compute *Self-BLEU* (Zhu et al., 2018) and *Distinct* (Li et al., 2016) scores for the set of selected samples. Table 5 shows *Matching rate* of the samples, Self-BLEU (quadrigram), and Distinct (bigram) scores of the matching samples for BB3-SL, STaR, and Hexa on seen KGD tasks. The result shows that Hexa produces most matching answers and achieves better performance in terms of diversity, indicating the capability of producing more diverse correct responses.

Table 5: Comparison of correctness and diversity of final responses between finetuning methods.

|         | Matching rate ↑ | Self-BLEU ↓ | Distinct ↑ |
|---------|-----------------|-------------|------------|
| BB3-SL  | 11.98           | 92.16       | 11.19      |
| STaR    | 12.92           | 92.75       | 10.81      |
| Hexa    | **13.98**       | **91.88**   | **11.51**  |

## 5.6 ABLATION STUDY

**Composing the Set** We hypothesize that the ideal set for formulating the guided prompt would be various candidates for the response. Our current design achieves this by using recent unmatched responses produced by the model, as they would become closer to the given ground truth as the learning progresses while keeping the variety. In order to investigate the hypothesis, we design an alternative method for composing the guided prompt for $x_i$ with the ground truth label of randomly selected $x_j$ where $j \neq i$. The results in Table 6 show that adding random ground truth of other samples in the set does not lead to any improvement compared to adding just the ground truth label. Furthermore, to show the effectiveness of using recent unmatched responses, we adversarially test by intentionally

Table 6: Comparing the effects of adding different guided prompts. The average scores across all tasks are shown. Super-scripted by * as the default setting for Hexa.

| Model | Seen | | Unseen | |
|-------|------|------|--------|------|
|       | F1   | R-L  | F1     | R-L  |
| BB3   | 17.83 | 19.48 | 15.91 | 14.29 |
| BB3-SL | 18.87 | 20.03 | 15.4 | 13.78 |
| STaR w/o hint | 18.74 | 21.96 | 16.06 | 14.73 |
| STaR | 19.25 | 20.84 | 16.4 | 15.03 |
| Hexa w/ random responses | 19.24 | 20.83 | 16.52 | 15.17 |
| Hexa w/o ground truth | 19.98 | **22.91** | 17.69 | 15.93 |
| Hexa w/ dissimilar responses | | | | |
| ∟ $K = 5, H = 4$ | 20.17 | 21.59 | 17.69 | 15.93 |
| Hexa w/ similar responses* | | | | |
| ∟ $K = 1, H = 4$ | 19.89 | 22.09 | 17.76 | 16.01 |
| ∟ $K = 5, H = 4^*$ | **20.83** | 22.25 | **18.55** | **16.63** |
| ∟ $K = 5, H = 1$ | 19.75 | 21.87 | 17.75 | 16.01 |

selecting a response with lowest similarity score with the ground truth among the $K$ samples when bootstrapping. The results are also shown in Table 6 with the label *Hexa w/ dissimilar responses*. Similarly, the results show that selecting the response with lowest score is detrimental to the performance. In addition, to test the effect of the response candidate size $K$ of Hexa (line 8 in Algorithm1), we run Hexa on two different values of $K$. The results are shown in the rows under the label *Hexa w/ similar responses* (see $K = 1, H = 4$ and $K = 5, H = 4$ in Table 6), and the use of the multiple response candidates helps the improvements.

When STaR does not use any additional guided prompts (denoted as *STaR w/o hint*), the learning objective is equivalent to policy gradient where the reward function is defined as exact matching function and it can be considered as the latent variable model optimized by RL. As presented in Table 6, STaR w/o hint shows better scores compared to BB3, slightly lower scores compared to STaR, and much lower scores than Hexa. To further clarify, we additionally investigate the performance when varying the number of unmatched responses $H$ to be included in the guided prompt in Hexa, which can be seen as a gradual transformation from STaR to Hexa. As shown in Table 6, we find that when Hexa includes only one latest unmatched response ($H = 1$) and the ground truth with the guided prompt, it outperforms STaR which only equips the ground truth with the guided prompt ($H = 0$), but underperforms Hexa's default setting ($H = 4$). Another point in the design of Hexa is the role of the ground truth. To test the effect of including the ground truth within the set, we ran an experiment where the guided prompt is composed without the ground truth, including only the previous response set. The result of this run is shown in Table 6 with the label *Hexa w/o ground truth* and also shows that it degrades performances when the ground truth is excluded. Interestingly, *Hexa w/o ground truth*, adding the falsely predicted responses only to the response set, performs better than *Hexa w/ random responses* and two version of *STaR*. It obviously shows that the self-generated responses can be meaningful information for the intermediate steps and response generation.

**Effect of Prompt Format**    Hexa augments the guided prompt to include the unmatched response set along with the ground truth in AL format without any extraneous prefixes. We design the prompt format to convey the intuition that the guided prompt set includes the ground truth, which is necessarily relevant information, and falsely predicted answers that may or may not be relevant information. Based on this assumption, we expect AL to function as a general form since it imposes autonomy to the model to interpret the guided prompt. To show this, we compare AL to *Bulleted List* formatting with bullet point (-) and AL with the prefix `Answer Choices:`. Here, the first implies a neutral set of all relevant information and the latter implies picking out a single relevant information. The results in Table 7 that both cases degrades performances.

Table 7: Comparison between different prompt formatting. The average scores across all tasks are shown. Super-scripted by * as the default setting for Hexa.

| Model | Seen | | Unseen | |
|---|---|---|---|---|
| | F1 | R-L | F1 | R-L |
| Alphabetical List (AL)* | 20.83 | 22.25 | 18.55 | 16.63 |
| `Answer Choices: AL` | 19.88 | 22.45 | 17.55 | 16.08 |
| Bulleted List | 19.49 | 21.94 | 17.86 | 16.02 |

## 6   DISCUSSION AND LIMITATIONS

**Ideals of Hexa**    Through Hexa, we ultimately aim to expand the solution in the perspective of curriculum learning (Bengio et al., 2009). Let us denote all possible ground truth context and response pairs as $\mathcal{G} = \{x_i^G, y_i^G\}_{i=0}^{|\mathcal{G}|}$. Then $\mathcal{D} \subset \mathcal{G}$, where $\mathcal{D}$ is a dataset with single label. If we assume that we have an ideal indicator function that can distinguish any given pair as a member of $\mathcal{G}$ and with the right prompt, Hexa would iteratively discover new set of pairs in $\mathcal{G} \setminus \mathcal{D}$. Therefore, in an ideal case, Hexa would be automatically performing a curriculum learning as the entropy the distribution over $\mathcal{G}$ would be increasing as more bootstrapped data are discovered. The conceptual illustration of the process of training set expansion is included in Figure 4 of Appendix B. However, Hexa does not fully follow the ideal case as the similarity measure used in $B$ cannot distinguishing all ground truth labels. We leave the problem of closing the gap from the ideal as a future direction of this work.

**Hexa as Reinforcement Learning**    As noted in Equation 3, the main objective of Hexa is very closely related to that of policy gradient method. The reward function can be of any form and thus the similarity score-based indicator function is still a valid reward function. However, there is an off-policy problem between caused by the difference between $p(\cdot|x_i, h_i^t; \theta)$ and $p(\cdot|x_i; \theta)$. A straightforward solution may be to apply importance sampling. By doing so, the newly formed objective would be more aligned with the primary objective in Equation 1. We leave adoption of off-policy correction techniques in reinforcement learning as possible expansion of this work.

**Hexa with LLMs**    We note there is no guarantee that our results would generalize to Large LMs (LLM). However, recent works (Schick et al., 2023; Li et al., 2023) that have similar process of one or few iterations of Hexa, namely finetuning on bootstrapped samples using standard MLE, suggest that the advantages of fine-tuning on augmented data persist at scale across various problems. This is expected if the additional training samples are beneficial to the problems as well as do not degrade the model's own capabilities. We also observe that OPT-175B (Zhang et al., 2022) significantly underperforms compared to BB3 models (e.g., 3B or 175B) on open-domain task under zero- and few-shot setting, as demonstrated in Shuster et al. (2022b). Furthermore, the performance gap between BB3-3B and BB3-175B is not very significant, which suggests that the most off-the-shelf datasets built for modular supervision may have limitations in enhancing the LLM-based modular systems. Therefore, they are still subject to failed responses, leaving room for improvement by Hexa. We leave this investigation to our future work.

## 7   CONCLUSION

We propose a novel self-improving method for open-domain, knowledge-grounded dialogue models that systematically generates dialogue responses using multiple intermediate modules. Specifically, we formulate the self-improving method with a bootstrapping scheme that uses a guided prompt for the model to produce suitable and diverse intermediate as well as final responses to be used for self-training. Experimental results demonstrate that the proposed method significantly outperforms the supervised learning and previous self-improving methods on various dialogue generation tasks.

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

# A   ALGORITHM

---

**Algorithm 1** Hexa algorithm

---

1: **Input:** $M$: model with parameter $\theta$, dataset $\mathcal{D} = \{x_i, y_i\}_{i=1}^{|\mathcal{D}|}$
2: **Initialize** response set $\{h_i^t\}_{i=1}^{|\mathcal{D}|}$ where $h_i^t = \{y_i\}$ and $t$ is the iteration number
3: **procedure** SELF-IMPROVING ITERATION
4:     **Initialize** bootstrap data $\hat{\mathcal{D}} = \{\}$
5:     **for** $n$ in $1...N$ **do**
6:         Sample $i \sim 1, ..., |\mathcal{D}|$                                   $\triangleright$ Sample datapoint
7:         $z \leftarrow M(x_i; \theta_t)$
8:         $y \leftarrow \mathrm{argmax}_{y_k} \mathrm{sim}(y_k, y_i)$, where $y_k \in \{M(x_i, z; \theta_t)\}_{k=1}^{K}$ $\triangleright$ Choose from $K$ responses
9:         **if** $B(y, y_i)$ **then**                        $\triangleright$ Match function by similarity score
10:             $\hat{\mathcal{D}} \leftarrow \hat{\mathcal{D}} \cup (x_i, z, y)$                    $\triangleright$ Bootstrapping data
11:         **else**
12:             $h_i^t \leftarrow h_i^t \cup y$                         $\triangleright$ Response set expansion
13:             $\hat{y} \leftarrow M(x_i, z, h_i^t; \theta_t)$              $\triangleright$ Guided prompt augmentation
14:             **if** $B(\hat{y}, y_i)$ **then**
15:                 $\hat{\mathcal{D}} \leftarrow \hat{\mathcal{D}} \cup (x_i, z, \hat{y})$             $\triangleright$ Bootstrapping data
16:             **end if**
17:         **end if**
18:     **end for**
19:     **for** mini-batch $d \subset \hat{\mathcal{D}}$ **do**
20:         $\theta_{t+1} \leftarrow \theta_t + \eta \nabla_{\theta_t} \mathrm{CE}(d)$         $\triangleright$ Model finetune using cross-entropy loss
21:     **end for**
22: **end procedure**
23: **repeat** Self-Improving Iteration until the performance plateaus

---

# B   ADDITIONAL ILLUSTRATIVE EXAMPLES

As supplementary illustrations of the processes described in the main paper, we show three figures in this section: 1) illustration of the process of search related branch of inference in Figure 2, 2) a graphical model of all branches of the intermediate steps in Figure 3, and 3) a conceptual illustration of curricullum learning described in section 6 in Figure 4.

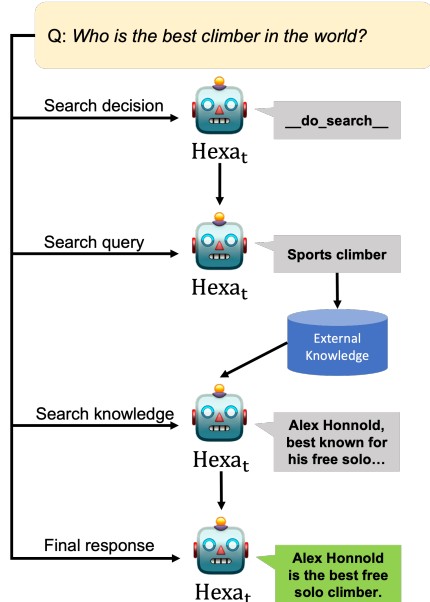

Figure 2: Example of external knowledge-grounded inference of our model. Here, we show an illustrative example of how the model inferences intermediate steps for external knowledge-grounded dialogue response generation. Following the same scheme as BB3 (Shuster et al., 2022b), given an input context, with a special token __is-search-required__, the model decides whether to search or not by outputting __do-search__ or __do-not-search__. Upon deciding to search, the model then generates a search query that will be used in the external knowledge source such as web, to retrieve relevant documents. For the query generation, a special token of __generate-query__ is appended at the end of the original context. With the retrieved documents, the model then generates a knowledge piece for the context using a special token __generate-knowledge__. Finally, with the generated knowledge appended to the context, the model generates the response for the given context.

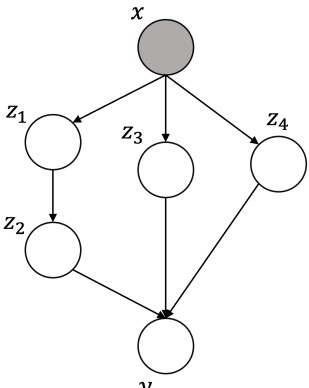

Figure 3: Graphical model of latent variables. Given the dialogue context $x$, $z_1 \sim p(\cdot|x; \theta)$ and $z_2 \sim p(\cdot|x, z_1; \theta)$ are the search query and the search knowledge respectively, where the search query is used as a query to retrieve external knowledge from sources such as web and the search knowledge is generated based on the retrieved external knowledge and $x$. $z_3 \sim p(\cdot|x; \theta)$ is the entity knowledge, generated using only the dialogue context $x$. Finally, $z_4 \sim p(\cdot|x; \theta)$ is the retrieved dialogue history-based internal knowledge, conditioned on $x$. After generating these intermediate steps, the final response $y \sim p(\cdot|x, z_{2:4}; \theta)$ is conditionally generated.

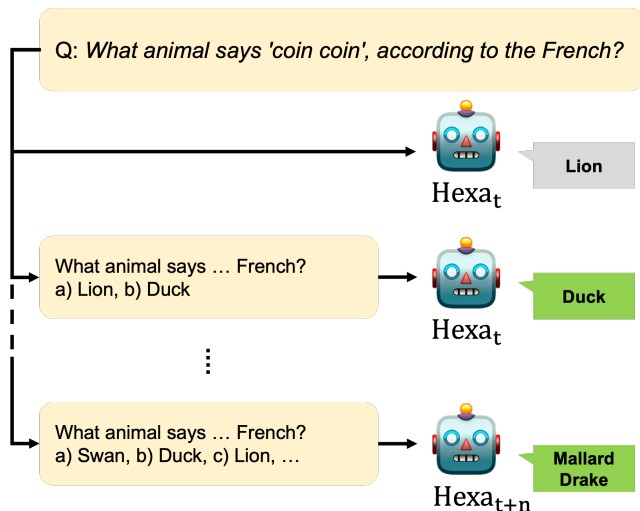

Figure 4: Conceptual illustration of curriculum learning in Hexa. Here, a question of *What animal says 'coin coin', according to the French?* with the ground truth *Duck* is given. The model at $t$ produces a wrong response *Lion* but attempts again with a guided prompt $h_i^t = \{\text{Duck}, \text{Lion}\}$, and the response is correct. After $n$ iterations, the model is asked with the same question again with a expanded set $h_i^{t+n} = \{\text{Swan}, \text{Duck}, \text{Lion}, ...\}$ and outputs *Mallard Drake*. Since *Mallard Drake* is a species of Duck, it can also be one of the ground truth output, and Hexa includes it in the training set.

## C    Module-wise Evaluation

As described in subsection 5.3 of the main paper, we report the module-wise evaluation for the instances where different models share the same decision paths. Note that memory-related modules used in multi-turn conversation scenarios were excluded from this experiment since it is impossible to compare the results under same condition, i.e., using exactly same memory of the conversation. The results in Table 8 show Hexa achieving the highest scores in search and entity knowledge generation. Combining this result with that of Table 1, we may draw an hypothesis that the performance in search and entity knowledge generation has relatively higher correlation with the performance of final response generation compared to the other two.

Table 8: Module-wise evaluation.

| Model | Search Query R-L | Search Knowledge R-L | Entity Knowledge R-L | Search Decision Accuracy |
|---|---|---|---|---|
| BB3 | 51.76 | 25.11 | 13.26 | 76.84 |
| BB3-SL | **52.40** | 21.90 | 20.07 | **79.52** |
| STaR | 48.88 | 24.81 | 22.17 | 77.76 |
| Hexa | 46.41 | **26.18** | **24.35** | 77.99 |

## D    Analysis on Bootstrap

### D.1    Bootstrap Samples

As mentioned in section 4 of the main paper, composing the guided prompt only with the ground truth, as in STaR, may collapse to simply copying the response throughout the intermediate steps, which can degenerate the generalization ability of the model. In order to empirically present such phenomenon, we compare the generated samples of search query and knowledge generation modules among different methods. We specifically analyze the search query samples from TriviaQA (Joshi et al., 2017) as this task is knowledge intensive QA with short responses where the copying phenomenon should be more easily observable. We report the rate of the number of search queries that include copies of the ground truth in bootstrapped samples with the guided prompt. As shown in the row labeled with *Search query copy rate* of Table 9, STaR's copy rate of search query is approximately twice the value of Hexa. Similarly, we also report the average overlap score by ROUGE-L between the generated knowledge and the ground truth in the bootstrap samples on all tasks except TriviaQA. As shown in the row labeled with *Knowledge overlap score* of Table 9, Hexa generates knowledge more dissimilar to the ground truth compared to STaR.

Table 9: The *Search query copy rate* (%) and the *Knowledge overlap score*. The average values across all iterations are presented.

| | STaR | STaR w/o hint | Hexa |
|---|---|---|---|
| Search query copy rate | 18.42 | 5.37 | 9.24 |
| Knowledge overlap score | 15.33 | 10.94 | 11.47 |

We observe that a variant of STaR that does not use the guided prompt, labeled with *STaR w/o hint* in Table 9 is inferior to Hexa even though it has lower values on both the copy rate and the overlap score than Hexa (see Table 6 of the main paper). This implies that reducing the copy rate or the overlap score may not be the direct cause of the improvement and the falsely predicted responses in the guided prompt of Hexa may make the bootstrap better than STaR. We hypothesize that the guided prompt of Hexa can provide reusable knowledge for the model to generate more informative response to the given context.

Although a generated response may not pass the matching function threshold, parts of it can contain information relevant to answering the given question. When this falsely predicted response is augmented in the guided prompt along with the ground truth, the guided prompt as a whole contains more relevant knowledge than just the ground truth, leading to generating more informative responses.

An example of such case is presented in Table 10 where the information from the falsely predicted response is combined with the ground truth in the final response.

Table 10: A bootstrap sample of Hexa on FITS Xu et al. (2022b). In this case, the first response is classified to the falsely predicted response (**Middle**). Then, Hexa tries to produce the response again along with the guided prompt that contains ground truth and the 1st response. In this 2nd try, the model generates the copy of ground truth into the knowledge. Along with the knowledge, the model generates the response which is classified to correct response and bootstrapped. The 2nd response also contains some part of the 1st response highlighted in olive (**Bottom**).

| | |
|---|---|
| **Input** | I'm trying to learn about sea turtles, could you help me find a list of places around the world where they live? |
| **Ground truth** | Green sea turtles can be found in more than 140 countries worldwide while the nesting grounds are found in 80 countries. |
| **1st response** (Falsely predicted) | Sea turtles are found all around the globe. Among the most common places they are listed are Belize, where they can be found on a regular basis, and the southern U.S./Southwest Mexico coast. They can still be seen in the Gulf as well, including florida, alaska and flores islands |
| Query | List of sea turtle habitat |
| Knowledge | Green sea turtles are found in more 140 countries worldwide while the nesting grounds are located in 80 countries |
| **2nd response** (Bootstrapped) | Green-sea turtle can be seen near the gulf of mexico and you can still find blue turtles in the southeast. They have found them in 80 countries, including florida, alaska and flores islands |

## D.2 BOOTSTRAPPING RATE

The bootstrapping rate, the number of bootstrapped data divided by the number of attempted instances, will be different for models depending on the used guided prompts. For example, STaR w/o hint, a version of STaR that does not take any guided prompt, may have a lower bootstrapping rate since it could be difficult to generate response similar to the ground truth without the guidance. To verify, we obtain the average bootstrapping rate across the iterations for different models with different guided prompts and the results are shown in Table 11. Interestingly, we find that Hexa has the highest bootstrapping rate, which greatly enhances the bootstrap data collection speed. More interestingly, Hexa w/o ground truth which only uses the unmatched responses for the guidance has better bootstrapping rate than STaR which only uses the ground truth for the guidance. This suggests that the self-generated responses are indeed meaningful information that correctly guides the response generation.

Table 11: Comparison for bootstrapping rate (%).

| | STaR | STaR w/o hint | Hexa | Hexa w/ random hint | Hexa w/o ground truth |
|---|---|---|---|---|---|
| Bootstrapping rate | 22.02 | 7.61 | **29.82** | 22.79 | 23.58 |

## E    EVALUATION ON TOTALLY-UNSEEN TASKS

Here, we provide an additional evaluation to test robustness of the methods. We consider OpenDialKG (Moon et al., 2019), a conversational reasoning benchmark dataset, consisting open-ended conversations between humans. In this task, the system is demanded to recommend items that users might prefer through multi-turn conversations on various domains including movies, books, sports, and music. Note that this task is not included during both BB3-training and finetuning (e.g. Hexa). As shown in Table 12, Hexa outperforms the other baselines in automatic evaluation for this totally-unseen task as well.

Table 12: Results for OpenDialKG unseen during BB3-training and finetuning.

| Model | F1 | R-L |
|---|---|---|
| BB3-60K | 15.02 | 13.84 |
| BB3-SL | 15.46 | 14.41 |
| STaR | 15.68 | 14.76 |
| Hexa | **18.08** | **16.60** |

## F   FURTHER DISCUSSION AND LIMITATIONS

**Bootstrap Quality**   An overlaying assumption in the self-improving methods such as Hexa is that samples with irrelevant $z$ would not be bootstrapped since they are unlikely to lead to appropriate responses. However, in practice, those cases may be included in the bootstrap and deteriorate the self-learning process. The current design does not include a mechanism to prevent this issue but a straight forward solution to such problem is to include a rejection sampling. For example, upon sampling an intermediate step $z$, we can decide to reject the sample if its presence and absence does not change the final response, meaning it has no relevance in producing the final response. This method can easily be extended to Hexa and we leave it as a possible candidate of future work direction.

## G   ADDITIONAL STUDY

### G.1   THRESHOLD SELECTION

Table 13: Different threshold values used for each task

| Task | Threshold |
|---|---|
| ***Question Answering*** | |
| TriviaQA Joshi et al. (2017) | 0.99 |
| MS Marco Nguyen et al. (2016) | 0.25 |
| ***Knowledge-Grounded Dialogue*** | |
| WoW Dinan et al. (2019) | 0.25 |
| WoI Komeili et al. (2022) | 0.25 |
| FITS Xu et al. (2022b) | 0.35 |
| ***Open-Domain Dialogue*** | |
| PersonaChat Zhang et al. (2018) | 0.35 |
| Multi-Session Chat Xu et al. (2022a) | 0.25 |
| ***Task-Oriented Dialogue*** | |
| GoogleSGD Lee et al. (2022) | 0.35 |

Before training, as described in subsection 5.2 of the main paper, we conduct a *task-wise threshold selection* that greedily searches the threshold value on each task to choose the appropriate threshold value, and use the selected threshold values (see Table 13) during training. We expect that this task-specific selection can lead to the performance improvement since undesired target responses can be bootstrapped when we inappropriately use a low threshold for the task, and only a narrow set of responses can be bootstrapped when we inappropriately use a high threshold for the task. To show these, we compare the performance between the *task-wise threshold* and *fixed threshold* $\in \{0.1, 0.25, 0.3, 0.4\}$ that uses the same threshold value for all tasks except TriviaQA (Joshi et al., 2017) which used the threshold value of $0.99$ as in the task-wise threshold. As shown in Table 14, the use of low threshold $b = 0.1$ degrades the overall performance while the use of high threshold $b = 0.4$ degrades the performance on unseen tasks. The use of median value $b = 0.25$ or approximate average value $b = 0.3 \approx 0.2929$ is inferior to the use of the task-wise threshold.

Table 14: Comparison between the *task-wise threshold* and *fixed threshold*.

|  | Seen | | Unseen | |
| --- | --- | --- | --- | --- |
| Threshold | F1 | R-L | F1 | R-L |
| *Task-wise* | **20.83** | **22.25** | **18.55** | **16.63** |
| *Fixed*, $b = 0.1$ | 18.69 | 21.43 | 16.88 | 15.28 |
| *Fixed*, $b = 0.25$ | 20.46 | 21.8 | 18.01 | 16.12 |
| *Fixed*, $b = 0.3$ | **20.8** | 22.07 | 18.16 | 16.24 |
| *Fixed*, $b = 0.4$ | 20.63 | **22.3** | 16.88 | 15.4 |

## G.2 MATCHING FUNCTION

As discussed in the main paper, the matching function $B$ could be essential in the proposed method. Here, we consider an alternative choice of similarity function called *Sentence-BERT* (S-BERT) (Reimers & Gurevych, 2019). Sentence-BERT was trained to measure the semantic similarity between two sentences and therefore can be used distinguish correct answers according to semantic similarity rather than the overlap. We replace ROUGE-L in matching function of Hexa with the cosine-similarity score between the S-BERT embeddings of the ground truth and the generated response. We label this setting as *Hexa w/ Sentence-BERT* in Table 15.

The results in Table 15 show that even with Sentence-BERT, Hexa achieves competitive scores in all metrics in both seen and unseen tasks, all higher than that of BB3-SL. Furthermore, we can observe that Hexa with ROUGE-L even improves in S-BERT score. Upon this observation, we conclude that ROUGE-L is effective and efficient choice of matching function, as S-BERT requires additional model inference to calculate the score.

Table 15: Ablation on similarity function. The Sentence-BERT score is denoted as S-BERT. Superscripted by * as the default setting for Hexa. The result of BB3 is added for reference.

|  | Seen | | | Unseen | | |
| --- | --- | --- | --- | --- | --- | --- |
| Model | F1 | R-L | S-BERT | F1 | R-L | S-BERT |
| BB3 | 17.83 | 19.48 | 41.37 | 15.91 | 14.29 | 34.37 |
| BB3-SL | 18.87 | 20.03 | 43.94 | 15.4 | 13.78 | 34.32 |
| Hexa w/ ROUGE-L* | **20.83** | **22.25** | 46.03 | **18.55** | **16.63** | **36.72** |
| Hexa w/ Sentence-BERT | 19.28 | 20.95 | **46.56** | 16.06 | 14.8 | 35.81 |

## H EVALUATION WITHOUT MODULES

Here, in order to examine the knowledge stored in the pre-trained model BB3 and trained model by Hexa, we evaluate the system to generate the final response directly, bypassing the intermediate steps before the final response generation. The average scores across the tasks categorized as Seen of the main paper are shown in Table 16 and we see that skipping the intermediate steps underperforms BB3 and Hexa with the intermediate steps.

Table 16: Evaluation models with (denoted as *w/*) and without (denoted as *w/o*) modules.

| Model | F1 | R-L |
| --- | --- | --- |
| BB3 w/o modules | 17.05 | 17.26 |
| BB3 w/ modules | 17.83 | 19.48 |
| Hexa w/o modules | 19.14 | 19.25 |
| Hexa w/ modules | 20.83 | 22.25 |

## I  HUMAN EVALUATION

To gauge performance across multiple aspects of quality of responses of dialogue generation by Hexa and STaR, we conduct the human evaluation on the nine tasks of KGD, ODD, and TOD. We randomly select 180 samples (20 samples per task), and each sample is evaluated by ten different human experts. Two responses with the generated knowledge from each model are shown to the annotators for each instance. For A/B testing on Fluency, Relevance, and Faithfulness, we give one score to the model if it's response is received an equally good or better than the other one, otherwise, we give zero score to the model. Figure 5 display the annotator instructions for the three aspects.

## J  REPRODUCIBILITY

Our code for Hexa is involved in the supplementary materials. Due to the file size limit, we include only the training and evaluation code that runs on a small subset of the original training and test data. We will release our entire code that runs on the full training and test data, and the trained model file upon acceptance.

Hexa is implemented in huggingface environment and the model is initialized with BB3-3B, which is publicly available. Hexa is trained using A100-80G 8GPUs for approximately 48 hours over 11 iterations.

Figure 5: Screenshot of one instance in the annotation tool used for human evaluation.

## K GENERATED SAMPLES

Table 17: Generated examples from WoW Dinan et al. (2019).

| Input | What is the most visited paris landmark? |
|---|---|
| query | Most visited place in paris |
| knowledge | Catacombs of Paris |
| **BB3-SL** | The catacombs are the most popular landmark in Paris. |
| query | Most visited place in paris |
| knowledge | Eiffel Tower |
| **STaR** | Eiffel tower. |
| query | Most visited place in paris |
| knowledge | The Eiffel Tower |
| **Hexa** | The eiffel tower is a landmark in paris. |
| **Input** | [...]
Yes, can you find out more about the very first ncaa football game, in 1869?
The first ncaa football game was played on nov 6, 1869, between rutgers and princeton. Rutgers won.
Great. One last question: what was the score of that game? |
| query | Rutgers v princeton ncaafb game score |
| knowledge | Rutgers won the game by a score of 6 runs to princeton's 4. |
| **BB3-SL** | Rutgers won that game by 6 runs.' |
| query | Rutgers princeton game score |
| knowledge | Rutgers won the game by a score of 6 runs to princeton's 4. |
| **STaR** | Rutgers won that game by 6 runs. |
| query | Rutgers princeton nca football game 1869 |
| knowledge | Rutgers won the game by a score of 6 to 4. |
| **Hexa** | Rutgers won their first game against princeton by a 6-4 score. |
| **Input** | What do they think jupiter's rocky core is made of? |
| query | Jupiter rocky core composition |
| knowledge | Gas giants consist mostly of hydrogen and helium. |
| **BB3-SL** | Jupiter is a gas giant, and gas giants consist of mostly hydrogen. |
| query | Jupiter rocky core |
| knowledge | Metallic hydrogen |
| **STaR** | Metallic hydrogen |
| query | Jupiter composition |
| knowledge | Jupiter's rocky core is made of liquid metallic hydrogen. |
| **Hexa** | Jupiter's core is believed to be made of metallic hydrogen |

Table 18: Generated examples from WoI Komeili et al. (2022).

| Input | Usually for 10-12 minutes, I like to get a little color on them. What is your favorite meal to make? |
|---|---|
| Input Persona | your persona: I like pasta. I know how to cook sausage. partner's persona: I enjoy cooking. I have a freezer. I like breakfast burrito. |
| query | pasta |
| knowledge | Banana pasta Banana pasta is a type of pasta prepared using banana as a primary ingredient. The product is typically made with unripe, [...] |
| BB3-SL | My favorite pasta is banana pasta. It is a pasta made with banana. |
| query | pasta |
| knowledge | Banana pasta is cooked like standard pasta, typically by boiling it. |
| STaR | My favorite pasta is banana pasta. It is cooked just like regular pasta. |
| query | pasta |
| knowledge | My favorite meal to cook is pasta. |
| Hexa | My favorite meal is pasta and I love to make it for my family. |
| Input | [...] Good for you! Are you watching the Superbowl this year? I will! It is on the 7th right ? It is. Are you a Brady fan or foe? |
| query | tom brady |
| knowledge | Tom Brady–Peyton Manning rivalry The Tom Brady–Peyton Manning rivalry is a series of games that took place between 2001 and 2016, involving two quarterbacks in the National Football League (NFL): [...] |
| BB3-SL | I am not a fan of Tom Brady, but I am a fan of Peyton Manning!' |
| query | tom brady |
| knowledge | The Tom Brady–Peyton Manning rivalry is a series of games that took place between 2001 and 2016, involving two quarterbacks in the National Football League (NFL): [...] |
| STaR | I am not a fan of Tom Brady but I am a fan of Peyton Manning |
| query | tom brady |
| knowledge | I am a fan of Tom Brady. |
| Hexa | I'm a big fan of tom brady! I like him a lot. I think he is a great quarterback. |

Table 19: Generated examples from MSC Xu et al. (2022a).

| | |
|---|---|
| **Input** | [...]
I just wanted something different at the moment! Would you ever consider dying your hair?
I don't think so, I'm way too old for that. I don't have much hair anymore!
Well, I think you could still do something cool perhaps! Any ideas? |
| Input Persona | your persona: I volunteer at a soup kitchen. I am depressed. I grew up in poverty. I am old.
partner's persona: I enjoy volunteer work. I like soup kitchens. I dyed my hair purple. |
| entity
**BB3-SL** | something
I'm not sure, maybe something with feathers? |
| entity
**STaR** | "
I have been thinking about getting a tattoo, but I'm not sure if I have the time or money for it. |
| entity
**Hexa** | the holidays
I would like to dye my hair pink for the holidays. I think it would be a good idea. |

| | |
|---|---|
| **Input** | [...]
Good, I was finally able to find a job that valued it. Who knew economics wouldn't be easily employable. What would you go back to school for?
Given your experience, possibly economics! Definitely something that doesn't involve hospitals.
Haha I can only imagine how bad you need a break from hospitals. If you'd like I can show you how to coupon to help save money when you go back in school |
| Input Persona | your persona: I have a job. I am considering going back to school to get a higher paying job. I am in school.
partner's persona: I have coupons. I won a coupon raffle. I have been to Disneyland. I have an economics degree. |
| entity
**BB3-SL** | money
That would be great! I'd love to learn how to save some money. |
| entity
**STaR** | economics
That would be great! I'd love to learn how to do that. I've never used coupons before. Do you have any tips for saving money in economics? |
| entity
**Hexa** | school
That would be great! I would love to learn how to do that. I would love to be able to save more money. I have been thinking about going to nursing school. |

Table 20: Generated examples from Ms Marco Nguyen et al. (2016).

| Input | why do electric dryers need to be vented |
|---|---|
| query | Electric dryer venting |
| knowledge | Hot dry(er) air enters the drum from either the front or the rear, and evaporates some of the moisture from the tumbling |
| **BB3-SL** | Hot dry air enters from the front and rear of the drum. |
| query | Electric dryer vent |
| knowledge | Hot dry(er) air enters the drum from either the front or the rear, and evaporates some of the moisture from the tumbling |
| **STaR** | Hot dry air enters from the front and rear of the dryer. |
| query | Electric dryer vent |
| knowledge | A clean, unobstructed dryer vent improves the safety and efficiency of the dryer. |
| **Hexa** | Dryer venting is important because it improves the efficiency and safety of dryers. |
| **Input** | amisom somalia |
| query | Buy amisom in somalia |
| knowledge | African Union Mission to Somalia (2007–present) |
| **BB3-SL** | African Union mission to Somalia |
| query | Amisom |
| knowledge | The African Union Mission to Somalia (AMISOM), guarding the palace, became involved in the fighting and was unable to take control of rebel strongholds. |
| **STaR** | The African Union mission to Somalia. |
| query | Somalia amisom |
| knowledge | The African Union Mission in Somalia (AMISOM) is an active, regional peacekeeping mission operated by the African Union with the approval of the |
| **Hexa** | The African Union mission in Somalia is a peacekeeping operation. |
| **Input** | why do puppies lick their owners |
| query | Puppy licking owner |
| knowledge | lick each other's faces in greeting, then they begin to sniff any moist membranes where odors are strongest i.e. mouth, nose, anal regions and urogenital areas. |
| **BB3-SL** | Puppies lick each other in greeting. |
| query | Puppy licking owner |
| knowledge | Licking can communicate information about dominance, intentions and state of mind |
| **STaR** | Licking is a way for puppies to communicate with their owners. |
| query | Puppy licking owner |
| knowledge | Licking can communicate information about dominance, intentions and state of mind, and like the yawn is mainly a pacifying behavior. |
| **Hexa** | Puppies lick their owner to communicate with them. Licking is a way for puppies to communicate with their owners. |

