# OpenReview forum: "Hexa: Self-Improving for Knowledge Augmented Dialogue System"
_ICLR.cc/2024/Conference — Submitted to ICLR 2024_

### Official Review · Reviewer_hXyG · 2023-10-30

**Soundness:** 3 good
**Presentation:** 2 fair
**Contribution:** 3 good
**Rating:** 5
**Confidence:** 4

**Summary:**

This work studies the knowledge-grounded dialogue response generation systems (KRGs). The authors assume that there are multiple intermediate steps before the final response generation.

The authors find that there is always no direct and suitable supervised data to train such intermediate modules. To this end, this work proposes a general self-improving framework Hexa, training intermediate modules without ground-truth labels.

Hexa framework first samples a bootstrap dataset even with the unmatched responses at each turn; then, Hexa tunes the model on the collected dataset to improve the performance.

**Strengths:**

1. Hexa has a good motivation.  KRG researchers are eager to see works that enhance the learning of intermediate modules.
2. The proposed Hexa can be used for other tasks that also have several intermediate modules/steps.
3. Multiple datasets are included for the empirical evaluation, and the proposed method mostly outperforms baselines, showing robust improvement.

**Weaknesses:**

1.  The authors have stated `2) a novel bootstrapping scheme with a guided prompt and a modified loss function for diverse and appropriate generation of intermediate and final responses to be self-trained;' as a major contribution. Nonetheless, there is no evaluation of the diversity in the main paper.

2.  Although Hexa has notable improvement in the evaluation of the generated dialogue responses, this work lacks enough experiments to verify *Does Hexa really improve the final performance by improving the performance of intermediate modules?*. I think this is important.

3. A large amount of text is piled up together in some paragraphs,  making this paper verbose and not easy to follow.

**Questions:**

N/A

---

> ### Author Response · Authors · 2023-11-23
>
> *We pay our gratitude to the reviewer for the time for the thorough review and the constructive feedbacks in the comments.*
>
> 1. **Format**: The authors have stated `2) a novel bootstrapping scheme with a guided prompt and a modified loss function for diverse and appropriate generation of intermediate and final responses to be self-trained;' as a major contribution. Nonetheless, there is no evaluation of the diversity in the main paper.
>
>     → Following your suggestion, we move the evaluation of the diversity from Appendix to main paper (see Section 5.5 in the revised paper).
>
> 2. **Importance of intermediate steps**: *Does Hexa really improve the final performance by improving the performance of intermediate modules?*
>
>     → In parts of the paper, we present evidence to the claim that the performance of final response if enhanced by improving the intermediate modules’ performances.
>
>     1. In Table 4, we show an example where a bad search query leads to retrieval of irrelevant documents, causing a generation of low quality response. Improving the intermediate modules help avoiding such cases.
>     2. In Table 16 of appendix, we show results comparing the performance of the responses with and without the intermediate modules. The results reveal a significant degradation in performance when intermediate modules are absent for both BB3 and Hexa.
>     3. In Table 8, we show results of module-wise evaluation of the intermediate modules. The result indicates that the final decision is correlated with the performance of the search and entity knowledge modules. The search decision and search query scores does not portray the relationship to the final response since the ground truth used to measure these score for the search query and decision may not be the optimal in our experiment setting due to the discrepancy in the search engine used.
>
>     We therefore conclude that enhancing the intermediate modules can lead to improvement in the final response for knowledge-augmented dialogue generation.
>
> 3. **Format**: A large amount of text is piled up together in some paragraphs, making this paper verbose and not easy to follow.
>
>     → Following your suggestion, we split overly long paragraphs to enhance verbosity and readability.

---

### Official Review · Reviewer_dRhK · 2023-11-01

**Soundness:** 3 good
**Presentation:** 3 good
**Contribution:** 3 good
**Rating:** 5
**Confidence:** 4

**Summary:**

This paper introduces Hexa, a self-improving modular mechanism based on a novel bootstrapping scheme with a guided prompt and a modified loss function. The data augmentation procedure involves including the past responses as part of a random Alphabetical list along with the GT input for the bootstrapping process. Empirical results on different knowledge tasks (question answering, knowledge-grounded dialogue, open-domain dialogue, and task-oriented dialogue) shows the efficacy of the proposed approach.

**Strengths:**

- The paper is well-motivated and clearly written in most parts.
- Ablation study for different components of the system is interesting and provides valuable insights into the effectiveness of the proposed approach.
- Human evaluation shows improvement over STaR on both training as well as held-out unseen dataset.

**Weaknesses:**

- It seems the relevant baselines haven’t been used (such as LLaMA 2-chat, GPT3/4) where the field has evolved a lot in recent years.
- The availability of code is not discussed. Implementation details related to the resources, framework, model cards, training days, etc. would help in reproducibility.
- Formatting of the paper could be improved with all the tables positioned closer to the text where they are referred in Section 5.3 and 5.4.
- It would have been interesting to see the performance of the model as the number of bootstrapped samples is linearly increased as briefly mentioned in Section 5.2.

**Questions:**

- Could the authors please clarify if the response was generated or ranked from a given list and how is the F1 score computed?
- Could the authors provide more detail about the experiments involving SentenceBERT for the similarity measure and also an intuition why it performed better/worse than BLEU/ROUGE?
- Could the authors provide more information about the human evaluation and if/how the inter-annotator agreement was computed?
- Could the authors explain how this approach would work in real-life scenarios?

Suggestions/Comments:
- It would help to specify BB3 as BlenderBot-3 when first introduced in Section 5.
- Section 5.5: Table 6 that -> Table 6 shows that
- Section 6: entropy the -> entropy of the

---

> ### Author Response · Authors · 2023-11-23
>
> *We pay our gratitude to the reviewer for the time for the thorough review and the constructive feedbacks in the comments.*
>
> 1. **Comparison for recent LLMs.**
>
>     → We do not consider models like LLama2-chat as a baseline for the following reasons:
>
>     1. Excessively large difference in the number of training tokens of the base model: The number of training tokens of LLama2 is approximately twenty times larger than the number of training tokens of BB3-3B (2T tokens for LLama2 vs 100B tokens for BB3).
>     2. Possibility of data contamination: Referencing scores from [1], we found that LLama2-7B-chat achieves unusually high score on the benchmarks used in this paper, e.g., TriviaQA, that is higher than the score of BLOOM-175B despite the size difference and the increase in the number of documents used by the BLOOM-175B model. This naturally causes a suspect that training data of LLama2 have high chance of being contaminated with the evaluation datasets used in this paper.
>
>     Please consider that with the growing concern of data contamination in LLM evaluations [2, 3], experimental design becomes more challenging when evaluating on popular benchmarks, as used in this paper. These benchmarks have not been explicitly defined as held-out datasets by the recent LLMs, such as LLama2. Additionally, since the training data is not publicly available, we cannot split the clean and contaminated evaluation sets, as done in [4].
>
>  `[1] Large Language Models Struggle to Learn Long-Tail Knowledge`
>
> `[2] Llama 2: Open Foundation and Fine-Tuned Chat Models`
>
> `[3] Rethinking Benchmark and Contamination for Language Models with Rephrased Samples`
>
> `[4] PaLM: Scaling Language Modeling with Pathways`
>
> 2. **Reproducibility.**
>
>     → Hexa is implemented in huggingface environment and the model is initialized with BB3-3B, which is publicly available. Hexa is trained using A100-80G 8GPUs for approximately 48 hours over 11 iterations. We include these statements in Appendix.J in the revised paper.
>
> 3. **Formatting.**
>
>     → The main text have been revised accordingly.
>
> 4. **Scoring.**
>
>     → In Hexa training, the responses are ranked by ROUGE-L from pairwise samples. In evaluation, we compute F1 values from all pairwise samples for a given list and choose the maximum value as the score.
>
> 5. **Detail about the experiments involving SentenceBERT.**
>
>     → Shown in Table 15 in the appendix, in the seen dataset, Hexa with ROUGE-L achieves highest ROUGE score while Hexa with sentenceBERT score as reward function achieves highest sentenceBERT score, as expected. The similarity measured by sentenceBERT behaves differently from the BLEU/ROUGE on the basis that BLEU/ROUGE is more strict while sentenceBERT may pick up non-overlapping similarity such as paraphrases and synonyms. Therefore, answers generated by Hexa w/ SentenceBERT may not necessarily have high overlapping scores like BLEU/ROUGE scores while responses generated by Hexa w/ ROUGE-L, based on word overlapping, will likely have high SentenceBERT score, as overlapping words would naturally lead to high semantic similarity.
>
> 6. **Inter-annotator agreement.**
>
>     → We provide the inter-annotator agreement below. We compute Cohen's Kappa scores for three aspects over Seen and Unseen splits. For each instance, each annotator assigned one of the four different labels (i.e., Both are good, Neither is good, A's response is good, and B's response is good). Then Cohen's Kappa scores was computed for all possible two-annotator combinations and the average scores are recorded in the table below. The results indicate a moderate agreement.
>
>     |  | Fluency | Relevance | Faithfulness |
>     | --- | --- | --- | --- |
>     | Seen | 30.95 | 42.95 | 50.88 |
>     | Unseen | 36.67 | 44.29 | 51.14 |
>
> 7. **Real-world scenarios.**
>
>     → We provide descriptions of real world application of Hexa in two different cases.
>
>     1. Cases that does not require training.
>
>         These are cases where the intermediate pipeline is identical or is a subset of the pipeline described in this paper. Further more, as long as the search engine operates similar to the ones used in our paper and the there is not much of a domain shift in the input queries, our trained model may be used straight out the the box, as seen in the performance of unseen-tasks in Table 12.
>
>     2. Cases that require re-training.
>
>         When there is a significant domain-shift of the use cases or when the pipeline differs from the one used in this paper, Hexa requires retraining of the model. In this case, the users are advised to collect input-output pairs and retrain Hexa. Note that since Hexa is a end-to-end training method, the users are not required to collect any of any training data of the intermediate modules beforehand. We leave further investigation of adapting Hexa to totally unseen pipeline as a future work.

---

### Official Review · Reviewer_uni2 · 2023-11-01

**Soundness:** 2 fair
**Presentation:** 2 fair
**Contribution:** 1 poor
**Rating:** 3
**Confidence:** 4

**Summary:**

In order to overcome the problem of dialogue systems missing clear intermediate data, the authors propose a self-improving modular approach that enhances both intermediate and final response generation through the use of a modified loss function and a guided prompt scheme. According to their empirical research, HEXA can generate conversation more effectively and outperform earlier techniques on a variety of dialogue tasks.

**Strengths:**

1. Self-Improving Mechanism: The method's ability to improve itself without ground truth data for intermediate steps is innovative and reduces dependency on large annotated datasets.

2. Empirical Performance: HEXA demonstrates superior performance on various benchmark datasets for knowledge-grounded dialogue tasks.

**Weaknesses:**

1. Metric Relevance: In the age of sophisticated dialogue systems like ChatGPT, conventional metrics like F1 and ROUGE-L might not be the most reliable measures of performance.

2. Lack of SOTA Comparison: The paper doesn't compare its method with state-of-the-art chat-based language models like LLama2-chat, which would be crucial for understanding its relative performance.

3. Outdated Baseline Model: The use of the STAR model as a baseline for human evaluation is noted as a potential weakness due to its age, which might not accurately reflect the current advancements in the field.

**Questions:**

Given that the paper does not include a comparison with state-of-the-art chat-based LLMs such as LLama2-chat, how do you anticipate HEXA would perform relative to these models? Additionally, could you discuss any plans to test HEXA's methodology on these larger models to validate its effectiveness in a more competitive landscape?

---

> ### Author Response · Authors · 2023-11-23
>
> *We pay our gratitude to the reviewer for the time for the thorough review and the constructive feedbacks in the comments.*
>
>
> 1. **Metric Relevance**: *In the age of sophisticated dialogue systems like ChatGPT, conventional metrics like F1 and ROUGE-L might not be the most reliable measures of performance.*
>
>     → We agree with the reviewer that according to the recent standards of characteristics that chat-models should have, such as helpfulness and insightfulness, measures like F1 and ROUGE-L may bot be the best option. However, we would like to also emphasize that in order to measure how well the response is grounded on the given knowledge, such as in the QA, KGD, and ToD tasks, metrics like F1 and ROUGE that measures the word overlap are reliable options and are also used in recent works like [1].
>
>     Also, there has been reports of biases that exists in the evaluation by LLMs such as GPT-4 [2]. On behalf of the matter, we provide results from more reliable metric of human evaluation in the paper, in which Hexa has shown a significant performance increase over the baseline.
>
>
> 2. **Lack of SOTA Comparison**: *The paper doesn't compare its method with state-of-the-art chat-based language models like LLama2-chat, which would be crucial for understanding its relative performance.*
>
>     → We do not consider models like LLama2-chat as a baseline for the following reasons:
>
>     1. Excessively large difference in the number of training tokens of the base model: The number of training tokens of LLama2 is approximately twenty times larger than the number of training tokens of BB3-3B (2T tokens for LLama2 vs 100B tokens for BB3).
>     2. Possibility of data contamination: Referencing scores from [3], we found that LLama2-7B-chat achieves unusually high score on the benchmarks used in this paper, e.g., TriviaQA, that is higher than the score of BLOOM-175B despite the size difference and the increase in the number of documents used by the BLOOM-175B model. This naturally causes a suspect that training data of LLama2 have high chance of being contaminated with the evaluation datasets used in this paper.
>
>     Please consider that with the growing concern of data contamination in LLM evaluations [4, 5], experimental design becomes more challenging when evaluating on popular benchmarks, as used in this paper. These benchmarks have not been explicitly defined as held-out datasets by the recent LLMs, such as LLama2. Additionally, since the training data is not publicly available, we cannot split the clean and contaminated evaluation sets, as done in [6].
>
>
> 3. **Outdated Baseline Model**: *The use of the STAR model as a baseline for human evaluation is noted as a potential weakness due to its age, which might not accurately reflect the current advancements in the field.*
>
>     → Of the recent works that bases on the scheme of bootstrap then fine-tune, [7, 8] may be a possible candidate baselines. However, the methodologies in these works are too specific to their applications, complicating the adaptation of their methods to knowledge-augmented dialogue generation. On the other hand, the method suggested in STaR is general enough that it can easily be adopted in knowledge-augmented dialogue generation task. Thus, we choose STaR as the baseline.
>
>
> 4. *Additionally, could you discuss any plans to test HEXA's methodology on these larger models to validate its effectiveness in a more competitive landscape?*
>
>     → As mentioned in section 6, we believe Hexa would similarly show improvements in knowledge augmented dialogues when applied with large models as it did with smaller ones. We leave this as a future work.
>
>
>
>
> `[1] Diverse and Faithful Knowledge-Grounded Dialogue Generation via Sequential Posterior Inference`
>
> `[2] Flask: Fine-grained language model evaluation based on alignment skill sets`
>
> `[3] Large Language Models Struggle to Learn Long-Tail Knowledge`
>
> `[4] Llama 2: Open Foundation and Fine-Tuned Chat Models`
>
> `[5] Rethinking Benchmark and Contamination for Language Models with Rephrased Samples`
>
> `[6] PaLM: Scaling Language Modeling with Pathways`
>
> `[7] Toolformer: Language models can teach themselves to use tools`
>
> `[8] Self-alignment with instruction backtranslation`

---

### Meta-Review · Area_Chair_7e7e · 2023-12-05

**Metareview:**

While the reviewers seemed to like the idea of the self-improving mechanism, they raised several concerns including lack of appropriate comparisons with state of the art methods, and lacking evaluation experiments to be convincing.

The authors justify why they did not use SOTA models (LlaMA 2-chat, GPT-X, etc) as having seen vastly larger amounts of data compared to the proposed model and high likelihood of data contamination. Given that only one reviewer acknowledged the authors' response I am less confident in my decision, but I do recommend to reject this paper.

**Justification For Why Not Higher Score:**

All reviewers raised similar concerns regarding baselines and evaluation and found the paper not convincing.

**Justification For Why Not Lower Score:**

N/A

---

### Decision · Program_Chairs · 2024-01-16

Reject